# Modulation of Heme-Induced Inflammation Using MicroRNA-Loaded Liposomes: Implications for Hemolytic Disorders Such as Malaria and Sickle Cell Disease

**DOI:** 10.3390/ijms242316934

**Published:** 2023-11-29

**Authors:** Alaijah Bashi, Cecilia Lekpor, Joshua L. Hood, Winston E. Thompson, Jonathan K. Stiles, Adel Driss

**Affiliations:** 1Department of Physiology, Morehouse School of Medicine, Atlanta, GA 30310, USA; abashi@msm.edu (A.B.); wthompson@msm.edu (W.E.T.); 2Department of Microbiology, Biochemistry, and Immunology, Morehouse School of Medicine, Atlanta, GA 30310, USA; cella20gh@yahoo.com (C.L.); jstiles@msm.edu (J.K.S.); 3Brown Cancer Center, School of Medicine, University of Louisville, Louisville, KY 40202, USA; joshua.hood@louisville.edu; 4Hepatobiology and Toxicology COBRE, University of Louisville, Louisville, KY 40202, USA; 5Department of Pharmacology & Toxicology, University of Louisville , Louisville, KY 40202, USA

**Keywords:** epigenetics, *Plasmodium falciparum*, malaria protection, exosomes, hemoglobinopathies, vaccine

## Abstract

Hemolytic disorders, like malaria and sickle cell disease (SCD), are responsible for significant mortality and morbidity rates globally, specifically in the Americas and Africa. In both malaria and SCD, red blood cell hemolysis leads to the release of a cytotoxic heme that triggers the expression of unique inflammatory profiles, which mediate the tissue damage and pathogenesis of both diseases. MicroRNAs (miRNAs), such as miR-451a and let-7i-5p, contribute to a reduction in the pro-inflammatory responses induced by circulating free hemes. MiR-451a targets both *IL-6R* (pro-inflammatory) and *14-3-3ζ* (anti-inflammatory), and when this miRNA is present, *IL-6R* is reduced and *14-3-3ζ* is increased. Let-7i-5p targets and reduces *TLR4*, which results in anti-inflammatory signaling. These gene targets regulate inflammation via *NFκB* regulation and increase anti-inflammatory signaling. Additionally, they indirectly regulate the expression of key heme scavengers, such as heme-oxygenase 1 (HO-1) (coded by the *HMOX1* gene) and *hemopexin*, to decrease circulating cytotoxic heme concentration. MiRNAs can be transported within extracellular vesicles (EVs), such as exosomes, offering insights into the mechanisms of mitigating heme-induced inflammation. We tested the hypothesis that miR-451a- or let-7i-5p-loaded artificial EVs (liposomes) will reduce heme-induced inflammation in brain vascular endothelial cells (HBEC-5i, ATCC: CRL-3245) and macrophages (THP-1, ATCC: TIB-202) in vitro. We completed arginase and nitric oxide assays to determine anti- and pro-inflammatory macrophage presence, respectively. We also assessed the gene expression of *IL-6R*, *TLR4*, *14-3-3ζ*, and *NFκB* by RT-qPCR for both cell lines. Our findings revealed that the exposure of HBEC-5i and THP-1 to liposomes loaded with miR-451a or let-7i-5p led to a reduced mRNA expression of *IL-6R*, *TLR4*, *14-3-3ζ*, and *NFκB* when treated with a heme. It also resulted in the increased expression of *HMOX1* and *hemopexin*. Finally, macrophages exhibited a tendency toward adopting an anti-inflammatory differentiation phenotype. These findings suggest that miRNA-loaded liposomes can modulate heme-induced inflammation and can be used to target specific cellular pathways, mediating inflammation common to hematological conditions, like malaria and SCD.

## 1. Introduction

Nearly half of the global population is at risk of developing malaria, a disease caused by the *Plasmodium falciparum* parasite that has claimed 619,000 lives [1]. The urgency of the situation is underscored by the fact that a child succumbs to this disease every two minutes, emphasizing the urgent need for effective interventions [1]. Malaria disease symptoms typically include fever and anemia due to the destruction of red blood cells (RBCs) [2]. The symptoms can range from mild complications to more severe forms, potentially resulting in organ damage, organ failure, or cerebral malaria (CM), which affects the brain [2]. In malaria, the breakdown of hemoglobin by the parasite releases free hemes [2]. Under normal conditions, heme scavenger proteins, like *hemopexin* and heme oxygenase-1 (HO-1) (coded by the *HMOX1* gene), bind hemes to prevent associated oxidative stress [3,4]. However, in cases of severe infection or high parasite load, excessive breakdown of infected RBCs (iRBCs) leads to the release of excessive hemes into the bloodstream and surrounding tissues. This overwhelms the protective capacity of *hemopexin*, causing the accumulation of free hemes in the bloodstream and tissues [5]. A free heme acts as a pro-oxidant, triggering oxidative stress and inflammation in various tissues [5]. Similar effects can be also observed in individuals with hemolytic disorders, like sickle cell disease (SCD) [6,7].

SCD is a genetic disorder caused by a mutation in the beta-chains of the hemoglobin (Hb) gene, resulting in morphological changes in RBCs under low-oxygen conditions (hypoxia) [6,8]. Notably, SCD has developed in malaria-prevalent areas as an evolutionary response to counteract the infection [9]. People with sickle cell trait (SCT, HbAS) have a lower malaria mortality rate due to *Plasmodium falciparum*’s reduced ability to replicate in their RBCs [10]. Sickle cell anemia (SCA, HbSS) occurs when both alleles of the mutated gene are inherited [11]. In both malaria and SCA, the breakdown of RBCs results in the release of a cytotoxic heme—a byproduct of hemoglobin—into the bloodstream [9,12]. This release triggers a range of complications, including vascular inflammation, oxidative tissue damage, and complications, such as cerebral malaria (CM) and stroke (in SCA) [5,13]. Pro-inflammatory cytokines are amplified, resulting in the chemotactic attraction of immune cells, like macrophages, to the vascular endothelial walls at sites of inflammation in blood vessels [14].

In the context of hemolytic disorders and heme-induced inflammation, macrophages play a pivotal role. They exhibit a pro-inflammatory profile that attracts leukocytes to sites of injury. This phenomenon also contributes to the promotion of angiogenesis and further underlines the critical role of macrophages in heme-induced inflammation [15,16]. Recent studies have shown that macrophages change phenotypically in response to their microenvironment [17]. Notably, two types of macrophages are of interest: the M1 type, associated with a killer response, and the M2 type, linked to a reparative response [17,18]. These macrophage types are characterized by specific markers, such as inducible nitric oxide synthase for M1 and type-I arginase for M2, both playing roles in l-arginine/nitric oxide metabolism [19,20]. The coexistence of M1 and M2 macrophages generates a robust immune response in heme-induced inflammation, such as in malaria and SCD [21]. 

There is a growing interest in identifying biomarkers to assess the progression of inflammation in the context of malaria and SCD pathogenesis [22,23,24]. MicroRNAs (miRNAs), a small non-coding RNA molecule, hold promise as biomarkers due to their involvement in gene regulation, their release during hemolytic diseases, and their potential to provide insights into disease processes [23,25,26]. Specifically, miR-451a, which targets the interleukin-6 receptor (*IL-6R*), and *14-3-3ζ* and let-7i-5p, which target toll-like receptor 4 (*TLR4*). Additionally, they indirectly modulate heme scavengers (*hemopexin* and *HMOX1*) which decrease cytotoxic heme concentrations [3,4,27]. *IL-6R*, *TLR4*, and *14-3-3ζ* all are involved in the modulation of the nuclear factor kappa B (*NFκB*) signaling pathway [27,28,29,30]. Additionally, the JAK/STAT3 and PI3K/AKT pathways mediate *IL-6R* and *TLR4* activation, further contributing to the organization of the inflammatory cascade [31,32]. These pathways play a crucial role in orchestrating the inflammatory response triggered by hemes and offer valuable insights into malaria and sickle cell disease pathogenesis [31]. 

MiRNAs can be transported in extracellular vesicles (EVs), including exosomes, which are small, membrane-bound vesicles. These vesicles facilitate the delivery of miRNAs to target cells, enabling them to mediate gene expression regulation [25,33,34]. In previous studies, we utilized artificial EVs, like liposomes, as carriers for miRNAs to investigate the potential connection between miR-451a and let-7i-5p and the pathogenicity of SCD and malaria [25].

Building on previous studies in our lab [25,35], we proposed a hypothesis that EV-loaded miRNAs can modulate heme-induced inflammation in macrophages and brain vascular endothelial cells.

To test our hypothesis, we conducted a series of in vitro experiments using liposomes carrying miR-451a or let-7i-5p mimic oligonucleotides on two cell lines: human brain vascular endothelial cells (HBECs-5i) and THP-1 macrophages. We utilized multiplexed immunoassays coupled with gene expression and arginase analysis to assess HBEC-5i and THP-1 cell responses to heme treatment when exposed to miR-451a or let-7i-5p mimic oligonucleotides. Our results showed that heme-induced inflammation in both cell types induced a pro-inflammatory response profile. However, administering liposomes loaded with miR-451a (Lipo-miR-451a) or let-7i-5p (Lipo-let-7i-5p) significantly reduced *IL-6R*, *TLR4*, and *NFκB* mRNA expression. Moreover, treating HBEC-5i cells with liposomes carrying let-7i-5p increased *HMOX1* and *hemopexin* expression, and treatment with both Lipo-miR-451a and Lipo-let-7i-5p significantly raised *14-3-3ζ* mRNA expression. We also noted a decrease in nitrite and an increase in l-arginine levels, indicating an anti-inflammatory macrophage response [20]. These results indicate that liposomes carrying miR-451a or let-7i-5p can effectively attenuate inflammation in vascular endothelial cells and macrophages. The potential therapeutic application of miRNA-loaded liposomes is evident, as they offer a promising approach to alleviate inflammation and cytotoxicity induced by hemes.

## 2. Results

### 2.1. Lipo-miR-451a Downregulates IL-6R mRNA in HBEC-5i and THP-1 Cells

In the study of HBEC-5i cells, the use of Lipo-miR-451a treatment has shown a significant reduction in *IL-6R* mRNA expression, regardless of whether the cells were treated with hemes or not, compared to the vehicle and heme-only treatments (Figure 1A). Similarly, treatment with Lipo-let-7i-5p significantly reduces *IL-6R* mRNA expression in the presence of hemes, but without hemes, the reduction is not significant compared to the DMSO (vehicle) treatment (Figure 1B).

In THP-1 cells, both Lipo-miR-451a and Lipo-let-7i-5p treatments have proven effective in significantly reducing *IL-6R* expression, regardless of heme induction (Figure 1C,D). Interestingly, cells treated with hemes alone exhibit a significant increase in *IL-6R* compared to all other treatments.

### 2.2. Lipo-let-7i-5 Downregulates TLR4 mRNA in HBEC-5i and THP-1 Cells

We also examined *TLR4* mRNA expression in HBEC-5i cells following treatment with Lipo-miR-451a or Lipo-let-7i-5p. The results revealed some interesting patterns. First, Lipo-miR-451a treatment led to a significant reduction in *TLR4* expression, both in the presence and absence of hemes (Figure 2A). In contrast, the results of the Lipo-let-7i-5p treatment were different. When a heme was present, *TLR4* expression significantly decreased; however, in the absence of a heme, there was no significant increase in *TLR4* expression (Figure 2B).

In THP-1 cells, we observed consistent results for both Lipo-miR-451a or Lipo-let-7i-5p treatments, showing a significant decrease in *TLR4* expression, regardless of heme induction (Figure 2C,D).

### 2.3. Lipo-miR-451a and Lipo-let-7i-5p Upregulates 14-3-3ζ in HBEC-5i Cells

We found that in HBEC-5i cells, *14-3-3ζ* significantly increased when treated with Lipo-miR-451a (Figure 3A) or Lipo-let-7i-5p (Figure 3B). Lipo-miR-451a significantly increased *14-3-3ζ* expression compared to heme-only treatment. Interestingly, when HBEC-5i cells were treated with Lipo-let-7i-5p alone, a significant decrease in *14-3-3ζ* expression was observed compared to treatment with Lipo-let-7i-5p (heme). However, compared to the heme-only treatment, we see a significant increase in *14-3-3ζ* expression in treatments with and without hemes.

### 2.4. Lipo-let-7i-5 and Lipo-miR-451a Downregulates p65/NFκB mRNA in HBEC-5i and THP-1 Cells

We investigated the relationship between Lipo-miR-451a and Lipo-let-7i-5p and their effects on *NFκB* mRNA expression by evaluating *p65*/*NFκB* expression.

In HBEC-5i cells, treatment with Lipo-miR-451a resulted in a significant reduction in *p65*/*NFκB* transcription, both in the presence and absence of hemes (Figure 4A). Lipo-let-7i-5p treatment led to a significant reduction in *p65*/*NFκB* mRNA expression only when a heme was present. Without hemes, there was no significant change (Figure 4B).

In THP-1 cells, when treated with Lipo-miR-451a, there was a decrease in *p65*/*NFκB* expression in the presence of hemes, but without heme co-administration, the change was not significant (Figure 4C). However, THP-1 cells treated with Lipo-let-7i-5p showed a significant reduction in *p65*/*NFκB* expression, regardless of heme treatment (Figure 4D).

### 2.5. Lipo-let-7i-5p Upregulates HMOX1 in HBEC-5i Cells

When we assessed *HMOX1* mRNA expression, we found an interesting trend. We found that there are only significant differences in HBEC-5i cells treated with Lipo-miR-451a (Figure 5A) and Lipo-let-7i-5p (Figure 5B). We see that compared to the heme-only treatment, there is a significant increase in *HMOX1* expression when treated with only Lipo-miR-451a. Furthermore, we see a significant increase in *HMOX1* expression when treated with Lipo-miR-451a with and without hemes compared to the vehicle-only treatment. Interestingly, in HBEC-5i cells treated with Lipo-let-7i-5p (Figure 5B), we see a significant increase in *HMOX1* expression in both the heme and no heme treatment groups. Given the trends that we see in *HMOX1* expression, we decided to look at *hemopexin*, another heme scavenger.

### 2.6. Lipo-let-7i-5p Upregulates Hemopexin in HBEC-5i and THP-1 Cells

The results of the analysis of *hemopexin* expression were remarkable given that we only found significant differences in HBEC-5i treated with Lipo-let-7i-5p (Figure 6B) and THP-1 treated with Lipo-let-7i-5p (Figure 6D). We observed a significant increase in *hemopexin* in HBEC-5i treated with Lipo-let7i-5p with and without hemes. We also observed a similar trend in THP-1 cells. Based on results from HBEC-5i cells and the trends observed in THP-1, we assessed the effects of our treatment on macrophages since they both play a role in exacerbating hemolytic disease severity. We determined the presence of nitrite in the macrophages to determine pro-inflammatory macrophage presence.

### 2.7. Arginase and Nitrite Concentration Is Decreased in THP-1 (Macrophages) Treated with miRNA

In THP-1-derived macrophages, we observed an increase in arginase activity when treated with Lipo-miR-451a. This observation is interesting, as increased arginase levels indicate a trend to an M2 anti-inflammatory phenotype. A similar tendency toward elevated arginase activity was observed in THP-1-derived macrophages exposed to Lipo-let-7i-5p treatment, albeit without reaching statistical significance (Figure 7B). We also detected a notable downward trend (without significance) in nitrite levels in THP-1-derived macrophages treated with both Lipo-miR-451a and Lipo-let-7i-5p (Figure 7C,D).

### 2.8. In Vitro Cell Death Assay

Figure 8 shows in vitro cell death assay data, and it appears that our model exhibits a greater degree of stability and lower toxicity in THP-1 macrophages compared to HBEC-5i endothelial cells.

## 3. Discussion

Both malaria and sickle cell diseases (SCDs) cause significant degradation of red blood cells (RBCs) and the subsequent release of cytotoxic hemes into the bloodstream [6]. Cytotoxic hemes can trigger inflammation and contribute to clinical symptoms [5,14,36,37]. Here, we investigated the role of heme-induced inflammation and the potential application of miRNA-loaded liposomes as a novel therapeutic strategy to mitigate the pro-inflammatory responses and cascades associated with cytotoxic hemes.

A cytotoxic heme triggers elevated levels of interleukin-6 (IL-6) and toll-like receptor 4 (*TLR4*), consequently activating signaling pathways such as JAK/STAT3, MYD88/IKK, and PI3K/AKT [28,29,38]. As a result, this cascade prompts *NFκB* activation and the generation of pro-inflammatory cytokines [31,32]. We hypothesized that treatment with liposome-loaded miRNAs could effectively mitigate the JAK/STAT3 and PI3K/AKT pathways [27,28]. Our results demonstrate the potential of Lipo-miR-451a and Lipo-let-7i-5p in regulating pro-inflammatory responses. These specific miRNAs exhibit the capacity to decrease the expression of *p65*/*NFκB* and levels of pro-inflammatory cytokines [25,28,32,37]. Our experimental data support the effectiveness of these treatments in both HBEC-5i and THP-1 cells.

In our study, we provide evidence that in both HBEC-5i and THP-1 cells, the application of Lipo-miR-451a or Lipo-let-7i-5p effectively reduces heme-induced *IL-6R* mRNA expression (Figure 1), even when it is not induced by hemes. These results have important implications for developing potential therapeutic applications of miRNA-based treatments in conditions involving *IL-6R* dysregulation.

We also show that both Lipo-miR-451a and Lipo-let-7i-5p can exert a significant impact on *TLR4* expression and are not dependent on heme induction (Figure 2). This indicates their potential to modulate the expression of pro-inflammatory cytokines, which may have important implications for managing inflammatory conditions.

Inflammatory conditions are regulated by a balance of anti- and pro-inflammatory cytokines and chemokines. *14-3-3ζ*, an important anti-inflammatory component, showed interesting results in our studies. We found a significant increase in *14-3-3ζ* expression in HBEC-5i cells in response to treatments with Lipo-miR-451a or Lipo-let-7i-5p (Figure 3). However, we see a decrease in *14-3-3ζ* expression in cells treated with let-7i-5p. To our knowledge, our study is the first of its kind to establish this relationship. *14-3-3ζ* is known to be involved in various cellular processes, including signal transduction and protein interactions, which intersect with pathways related to inflammation and immune responses through *NFκB* signaling. A higher expression of *14-3-3ζ* has been associated with M2 macrophage polarization [19,27,39,40]. 

*IL-6R*, *TLR4*, and *14-3-3ζ* are all known to impact the transcription of *P65*/*NFκB* directly and indirectly [27,28,29]. These facts are supported by our results in Figure 4, further confirming the role of Lipo-miR-451a and Lipo-let-7i-5p in regulating inflammatory responses and their potential application as a therapy for heme-induced inflammation.

Given the role of chronic inflammation due to cytotoxic hemes, heme scavengers, such as *HMOX1*, were considered potential targets for miRNA treatment. In HBEC-5i cells, we observed significant differences in response to treatments with Lipo-miR-451a or Lipo-let-7i-5p (Figure 5). Our experiments suggest that both Lipo-miR-451a and Lipo-let-7i-5p increase *HMOX1* expression in HBEC-5i cells (and not THP-1 cells), contributing to heme scavenging mechanisms in the presence or absence of hemes. The trends delineated in *HMOX1* expression create a link between miRNA treatments, hemes, and scavenging mechanisms. These findings prompt a broader investigation into the molecular mechanisms driving these effects and prompted us to delve into additional factors, like *hemopexin*, to attain a more comprehensive understanding of the intricate regulatory networks at play. Specifically, within HBEC-5i cells, treatment with Lipo-let-7i-5p resulted in a noteworthy increase in *hemopexin* expression, both in the presence and absence of hemes (Figure 6). However, we did not see this trend in HBEC-5i cells treated with Lipo-miR-451a. This observation was mirrored by similar trends identified in THP-1 cells. The consistency between these cell types enhances the credibility of our findings. However, it is possible that let-7i-5p may be more effective at increasing *hemopexin*, which would have a direct role in reducing excess hemes.

The insights drawn from our HBEC-5i findings, combined with the observed patterns in THP-1 cells, motivated us to further investigate the impacts of our treatments on macrophages. This choice was informed by the roles that both endothelial cells and macrophages play in exacerbating the severity of hemolytic diseases. To assess the presence of pro-inflammatory macrophages, we turned our attention to the evaluation of nitrite within these macrophage populations.

Our findings, as depicted in Figure 7, effectively demonstrate the potential utility of Lipo-miR-451a or Lipo-let-7i-5p in promoting the augmentation of M2 macrophage phenotypes. This holds a crucial role in the development of therapeutics applicable to diseases characterized by elevated levels of inflammation. Nevertheless, nitrite serves as an indicator of a pro-inflammatory or M1 macrophage phenotype. Also, in Figure 7, we successfully demonstrated an elevation in arginase levels alongside a reduction in nitrite, aligning with the characteristic pattern of diminished inflammation [19]. The secretion of nitrite is attributed to pro-inflammatory macrophages, signifying heightened inflammation [41]. In the context of the decrease in nitrite levels and the concurrent increase in L-arginine levels, this dual pattern assumes importance as an indicator of anti-inflammatory macrophages [19].

We explored the role of cytotoxic hemes in triggering chronic inflammation and further investigated the potential role of miRNA-loaded liposomes as a novel therapeutic strategy to mitigate pro-inflammatory responses caused by free hemes. Our results demonstrated that Lipo-miR-451a and Lipo-let-7i-5p effectively regulated pro-inflammatory pathways, reducing the expression of *p65*/*NFκB* and pro-inflammatory cytokines in both HBEC-5i and THP-1 cells while remaining relatively non-toxic to the cells (Figure 8). This suggests that our treatment can modulate these molecular pathways and produce an anti-inflammatory environment in those where cytotoxic hemes are present.

We propose a model that examines the molecular pathway encompassing *NFκB* transcription, *IL-6R*, and *TLR4* receptor activity, and supports our hypothesis (Figure 9). In the model, we show through the delivery of miR-451a and let-7i-5p via liposomes that we can reduce *NFκB* transcription, which effectively reduces inflammation. Upon the activation of *TLR4* and *IL-6R*, facilitated by the presence of hemes or IL-6, there is an elevation in *NFκB* transcription and subsequent production of pro-inflammatory cytokines. Conversely, when subjected to treatment with Lipo-miR-451a or Lipo-let-7i-5p, the expression of *IL-6R*, *TLR4*, and *NFκB* is downregulated. We propose that this cascade of events contributes to the attenuation of pro-inflammatory cytokine levels.

*NFκB* is a transcription factor that plays a key role in regulating the expression of various genes involved in the immune response, inflammation, cell survival, and other processes. *NFκB* can exhibit both anti-inflammatory and pro-inflammatory effects, depending on the context in which it is activated [42]. In the cases of SCD and malaria, elevated heme levels induce heightened *NFκB* activation, subsequently driving the production of downstream pro-inflammatory cytokines [31,37,43]. Excessive pro-inflammatory cytokines lead to chronic inflammation and ultimately the clinical manifestations associated with these diseases [7,37,44]. Here, we observed a pro-inflammatory phenotype for *NFκB* in the presence of excessive hemes. However, this inflammatory response can be mitigated by miRNAs miR-451a and let-7i-5p (Figure 9). This observation is significant in that it provides a promising opportunity wherein miR-451a and let-7i-5p could potentially be used to ameliorate the impact of chronic inflammation inherent in hemolytic diseases.

Recent studies have supported our hypothesis that miRNAs can attenuate inflammation in different cell types [45]. Our study elevates this idea by demonstrating how miR-451a or let-7i-5p liposomes can decrease pro-inflammatory responses in endothelial cells and macrophages (Figure 1, Figure 2 and Figure 4) and increase anti-inflammatory responses in endothelial cells (Figure 3). Furthermore, our investigation underscores the efficacy of Lipo-miR-451a or Lipo-let-7i-5p treatment in bolstering heme scavengers, like *HMOX1* (Figure 5) and *hemopexin* (Figure 6), within vascular endothelial cells. This study also illuminates our ability to modulate the balance of pro-inflammatory and anti-inflammatory macrophages through Lipo-miR-451a or Lipo-let-7i-5p treatment. Collectively, these findings strongly indicate that these specific miRNAs could serve as potent tools not only in managing malaria but also in addressing a spectrum of hemolytic diseases, including conditions like SCD. These dual potential underscores the promising implications of our research for broader clinical applications.

The relationship between malaria disease and sickle cell disease (SCD) is intricate. SCD, an inherited blood disorder characterized by abnormal hemoglobin and heightened cytotoxic hemes and inflammation, shares a complex interaction with malaria [24,46]. Interestingly, individuals possessing the sickle cell trait (SCT) exhibit increased resistance to *Plasmodium falciparum* infection and reduced symptom severity due to alterations in RBC morphology and elevated miRNA biomarkers [24,35]. Given this backdrop, our research carries significant implications for pioneering innovative approaches to address heme-induced inflammation in hemolytic disorders, notably SCD.

Although miRNAs miR-451a and let-7i-5p hold promise as potential therapeutic agents, a comprehensive exploration of their molecular impact on heme-induced inflammation and associated signaling pathways remains imperative. Elucidating their precise molecular mechanisms and assessing the effectiveness of liposomal encapsulation as a delivery method are pivotal steps to determine the viability of miRNAs as therapeutic interventions. The refinement of this strategy has the potential to yield clinical applications.

In this study, we did not combine the miRNAs into one liposome because we wanted to establish the effects of miR-451a and let-7i-5p individually. Future studies will focus on combining these two miRNAs into one liposome in different ratios to determine if there is an optimal combination to elicit the most therapeutic response. Future studies will also focus on let-7i-5p and the mechanisms associated with *hemopexin* expression and heme presence. With our results, we see that both miR-451a and let-7i-5p can effectively reduce pro-inflammatory gene expression and increase anti-inflammatory gene expression through different pathways. Determining whether Lipo-let-7i-5p or Lipo-miR-451a is better as a therapeutic option depends on the specific therapeutic goals and the context of the disease or condition being targeted. Lipo-let-7i-5p may be a better choice if the primary goal is to reduce heme-induced inflammation and enhance heme clearance, especially in situations where hemes are involved. On the other hand, Lipo-miR-451a appears to have a broader and more consistent anti-inflammatory effect, making it a strong candidate for conditions involving inflammation, both with and without hemes. The choice between the two should be based on the specific therapeutic objectives, taking into account the disease context and potential combination therapies. Further research and experimentation may also help in determining the most appropriate treatment option.

Moreover, the successful outcomes stemming from these investigations may lay the groundwork for tailored therapies in conditions characterized by heightened inflammation, including type 1 diabetes and autoimmune diseases, like lupus. These advancements could hold promise for individuals afflicted by such conditions [16]. Further research and optimization are essential to unlock the full potential of miRNA-based interventions and their broader applicability in tackling inflammatory diseases.

## 4. Materials and Methods

### 4.1. Liposome Formulation

Synthetically derived PEGylated liposomes were developed in the laboratory of Dr. J.L. Hood at the University of Louisville. These liposomes contained either a miR-451a mimic (ThermoFisher, cat# 4464066, Waltham, MA, USA) or let-7i-5p mimic (ThermoFisher, cat # 4464066, Waltham, MA USA), using an equal percentage of lipid fractions. Briefly, the following lipid co-mixture was solubilized in chloroform (Sigma-Aldrich, cat # CX1060-1; St. Louis, MO, USA): 64.89 mol% lecithin (Avanti Polar Lipids Inc., Alabaster, AL, USA, cat # 850705P-25mg), 32 mol% cholesterol (Sigma-Aldrich, cat # NC9138103), 65.99 mol% phosphatidylcholine (Avanti, cat # 840051P), 1 mol% 16:0 phosphatidylethanolamine (Avanti, cat # 850705P), 1 mol% 1,2-distearoyl-sn-glycero-3-phosphoethanolamine-N-[carboxy(polyethylene glycol (PEG))-2000] (Avanti, cat # 880135P), and 0.01 mol% 1,1′-Dioctadecyl-3,3,3′,3′-tetramethylindocarbocyanine perchlorate (Millipore Sigma, Burlington, MA, USA, cat # 42364-100MG). The mixture was then dried to a lipid film under a continuous vacuum using an IKA RV 10 rotary evaporator. The residual solvent was removed by drying under a continuous vacuum. The resulting dry lipid film was then resuspended in 1 mL of miR-451a or let-7i-5p mimic in DNAse/RNAse free, purified water. Liposomes were then formed using a QSonica Q500 sonicator. The sizes of the liposomes were determined by Dr. Ming Bo Huang in the Microvesicles Core Lab at the Morehouse School of Medicine using a NanoSight LM10-HSBF Nanoparticle Characterization System (Malvern Panalytical Ltd., Malvern, UK). Furthermore, the zeta potential was calculated by the J.L. Hood lab using a PMX 120 ZetaView^®^ (Particle Metrix Inc., Ammersee, Germany) Nanoparticle Tracking Analyzer. The methods employed for liposome preparation and analysis followed established protocols [47]. We named Lipo-miR-451a for the liposomes loaded with miR-451a and Lipo-let-7i-5p for the liposomes loaded with let-7i-5p. The final concentration of miRNA in liposomes was 400 pmol. The expression of miRNAs let-7i-5p and miR-451a was an influence on heme-induced inflammation [48].

### 4.2. Cell Culture

Human brain endothelial cells (HBEC-5i; ATCC cat # CRL-3245, Manassas, VA, USA) were initially authenticated and subsequently cultured according to the ATCC protocol. Briefly, cells were partially thawed and added to fresh DMEM: F12 media supplemented with 10% fetal bovine serum (FBS; and 40ug/mL endothelial growth supplement (ECGS) (Sigma Aldrich cat # E2759, St. Louis, MO, USA). Tissue culture plates were coated with a 0.1% gelatin solution for 1 h before the addition of HBEC-5i cells. Cells were maintained in culture on 0.1% gelatin (Stemcell Technologies; NC1620050, Cambridge, MA, USA) for one week and passaged every two days with fresh media until liposome administration. Incubation conditions were 37 °C and 5% CO_2_.

Monocytes (THP-1; ATCC cat # TIB-202) were initially authenticated and subsequently cultured and maintained for one week with fresh media every two days, according to the ATCC protocol, until ready for treatment. Briefly, cells were thawed and added to RPMI1640 supplemented with 10% FBS (Gibco; cat # 16-000-044) and 2-mercaptoethanol (final concentration of 0.05 mM). Cells were maintained as macrophages for 24 h, passaged as per the manufacturer’s protocol, and incubated at 37 °C with 5% CO_2_.

### 4.3. Macrophage Conversion

Once THP-1 cells reached confluency, cells were treated with 10ng/mL of Phorbol 12-myristate 13-acetate (PMA; ThermoFisher cat # J63916.MCR; Waltham, MA, USA) for 24 h to differentiate monocytes into macrophages [49].

### 4.4. Liposome Treatment

Both HBEC-5i cells and THP-1 cells were cultured in 24-well plates. For HBEC-5i, the plates were coated with 0.1% gelatin for 1 h prior to the addition of cells. Cells were added and allowed to proliferate until 80–100% confluency. Once cells reached confluency, media were replaced with DMEM: F12 complete media supplemented with 10% exosome-depleted FBS (Gibco; cat # A2720801; New York, NY, USA) and ECGS. HBEC-5i cells were then treated with either Lipo-miR-451a with and without 30 µM hemes (Sigma-Aldrich, cat # H9039; St. Louis, MO, USA), Lipo-let-7i-5p with and without 30 µM hemes (cat # H9039), or a volume -equivalent of a DMSO vehicle (ThermoFisher, cat # 85190; Waltham, MA, USA) for 24 h.

For THP-1, cells were added to the 24-well plate and allowed to proliferate until 80–100% confluency. Once THP-1 cells reached confluency, cells were treated with 10 ng/mL of Phorbol 12-myristate 13-acetate (PMA; ThermoFisher cat # J63916.MCR; Waltham, MA, USA) for 24 h to differentiate monocytes into macrophages [49]. A total of 24 h after PMA treatment, media were replaced with RPMI-1640 complete media supplemented with 10% exosome-depleted FBS and 2-mercaptoethanol. Cells were treated with either Lipo-miR-451a with and without 30 µM hemes (Sigma-Aldrich, cat # H9039; St. Louis, MO, USA), Lipo-let-7i-5p with and without 30 µM hemes (cat # H9039), or a volume -equivalent of a DMSO vehicle (ThermoFisher, cat # 85190; Waltham, MA, USA) for 24 h. We chose the precise concentration of 30 µM hemes based on our previous research, demonstrating that it mirrors plasma levels during malaria infection. Furthermore, our in vitro experiments have revealed that this heme concentration serves as the threshold for initiating inflammation without reaching toxic levels in a dose-specific study.

### 4.5. Total RNA Isolation and RT-qPCR

Total RNA was collected from HBEC-5i and THP-1 cell culture 24 h after liposome treatment. Total RNA was collected using a TRIzol LS Reagent (Invitrogen, cat # 15596026; Waltham, MA, USA), as per the manufacturer’s protocol, and was eluted in UltraPure™ DNase/RNase-Free Distilled Water (ThermoFisher cat # 10977015; Waltham, MA, USA). RNA concentration and purity were determined using a ThermoScientific NanoDrop 1000 Spectrophotometer. Samples were stored at −80 °C until later analysis. MiRNA was reverse transcribed using an iScript cDNA synthesis kit (Bio-Rad, cat # 1708890; Hercules, CA, USA), as per the manufacturer’s protocols. PCR was completed using SsoAdvanced Universal SYBR Green Supermix (Bio Rad, cat # 1725270; Hercules, CA, USA), and RT-qPCR was performed using Bio-Rad CFX Real-Time PCR Detection Systems. Primers for *IL-6R*, *TLR4*, *P65*/*NFκB*, and GAPDH (internal control) were purchased from Genewiz and are listed in Appendix A.

### 4.6. Nitrite Assay Analysis

Nitrite assay analysis was carried out using the manufacturer’s protocol (Sigma-Aldrich. cat# MAK367). Briefly, cell lysate from a THP-1 cell culture with liposome treatment was homogenized with a cold nitrite assay buffer. Samples were chilled on ice for 10 min followed by centrifugation at 10,000× *g* for 4 min. The supernatant was collected and added to a 96-well plate for analysis. The remaining nitrite assay buffer was added to each sample to reach 100 µL. Standards were created using the manufacturer’s protocol. The plate was allowed to incubate at room temperature for 10 min. After the incubation period, the plate was read at 540 nm. The standard curve was calculated, and ODs were used in statistical analysis.

### 4.7. Arginase Assay

Arginase assays and calculations were conducted using the manufacturer’s protocol (Sigma-Aldrich. cat# MAK112). Briefly, cell lysate was centrifuged at 1000× *g* for 10 min. Cells were lysed using 10 mM of Tris-HCl and were centrifuged for 13,000× *g* for 10 min. The supernatant was collected and brought to a final volume in a 96-well plate. A substrate buffer was added to each sample and allowed to incubate at 37 °C for 2 h. After incubation, an urea reagent was added to stop arginase reactions. Absorbance was read on a microplate reader at 430 nm. Arginase activity was calculated according to the manufacturer’s equations and instructions.

### 4.8. Statistical Analysis

Statistical analysis was completed using GraphPad Prism version 10.0.0 for Windows (GraphPad Software, La Jolla, CA, USA). The ΔΔCT method was used to identify mRNA gene expression [48]. Normality was checked using a Shapiro-Wilk normality test. The ΔΔCTs of the expressions of *IL-6R*, *TLR4 P65*/*NFκB*, and GAPDH were calculated using Microsoft Excel (Office 365; Version 2302). One-way ANOVA and Tukey’s multiple comparison tests were used to test for significance between groups. If the data were not normal, a Kruskal-Wallis test was used. A significant *p*-value of <0.05 was set for all analyses.

## Figures and Tables

**Figure 1 ijms-24-16934-f001:**
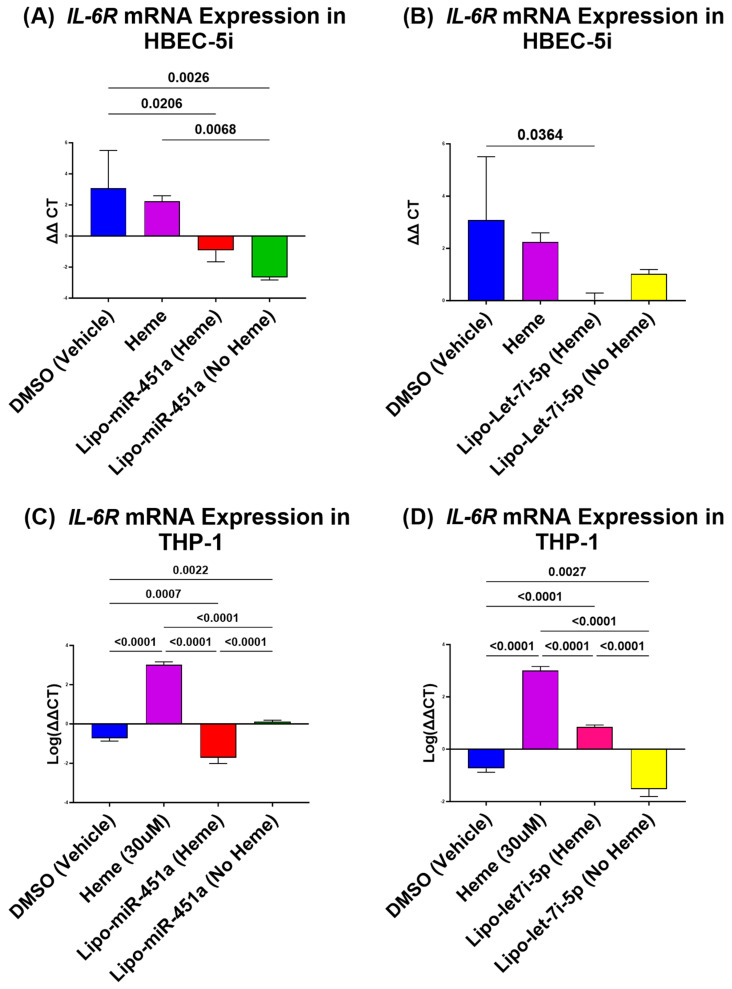
*IL-6R* mRNA expression in HBEC-5i (**A**,**B**) and THP-1 (**C**,**D**) cells. (**A**) Lipo-miR-451a treatment: *IL-6R* expression is significantly reduced in cells treated with or without hemes compared to vehicles and hemes only. (**B**) Lipo-let-7i-5p treatment: *IL-6R* significantly reduced with hemes but not significant without hemes compared to vehicles. (**C**) Lipo-miR-451a treatment: cells with and without heme treatment have a significant reduction in *IL-6R* both compared to those with heme treatment-only and vehicles. Heme-only-treated cells have a significant increase in *IL-6R* compared to all other treatments. (**D**) Lipo-let-7i-5p treatment: there is a significant decrease in *IL-6R* expression with and without heme treatment. A significant increase was found in those treated with hemes only compared to other treatments. These results show that both Lipo-miR-451a and Lipo-let-7i-5p decrease heme-induced *IL-6R* gene activity in both cells. *IL-6R* expression is also modulated by lipo-miRNAs when not induced by hemes.

**Figure 2 ijms-24-16934-f002:**
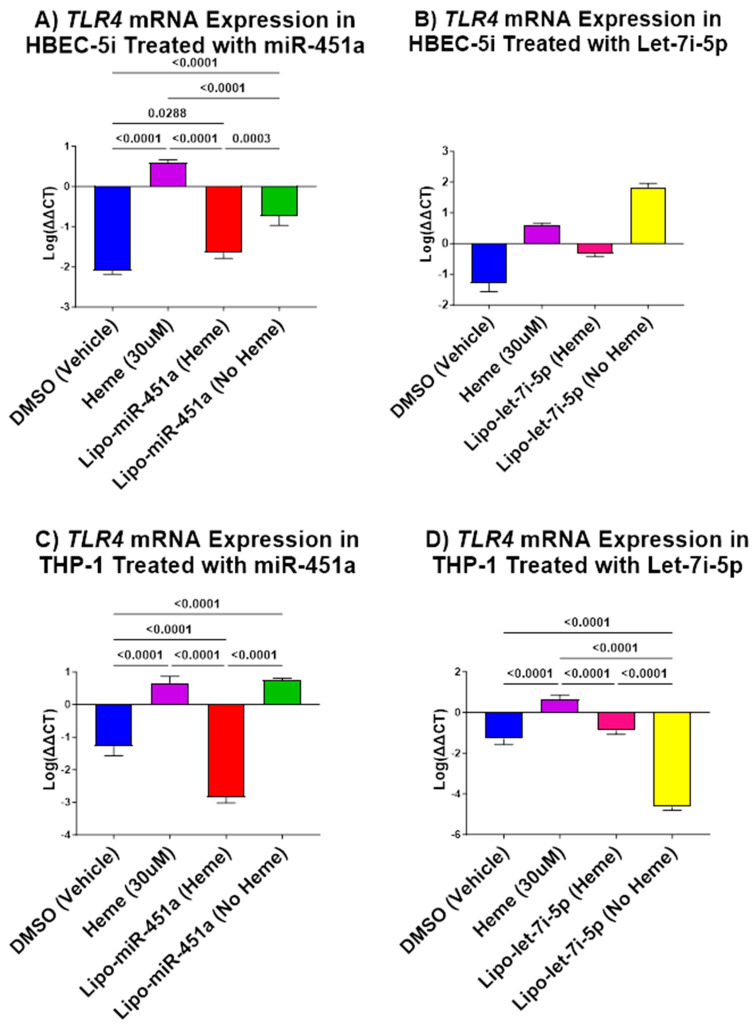
*TLR4* mRNA expression in HBEC-5i (**A**,**B**) and THP-1 (**C**,**D**) cell culture. (**A**) Lipo-miR-451a: *TLR4* significantly reduced when treated with and without hemes compared to both heme- and vehicle-only treatments. (**B**) Lipo-let-7i-5p: Compared to heme-only and vehicle-only treatment, cells treated with hemes and liposomes significantly reduce *TLR4*, whereas no heme significantly increases *TLR4*. (**C**) Lipo-miR-451a: *TLR4* significantly reduced when treated with hemes and liposomes and increased without hemes but with liposome treatment. (**D**) Lipo-let-7i-5p: *TLR4* expression significantly decreased when treated with liposomes with and without hemes. These results show that both Lipo-miR-451a and Lipo-let-7i-5p have a significant impact on *TLR4* expression, both with and without hemes. Our treatment has been shown to reduce the expression of *TLR4* and, by association, the expression of pro-inflammatory cytokines.

**Figure 3 ijms-24-16934-f003:**
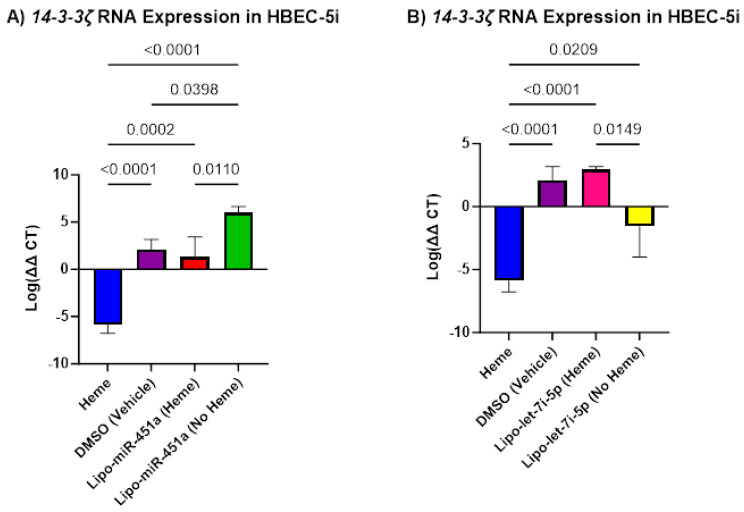
*14-3-3ζ* mRNA expression in HBEC-5i cell culture. (**A**) Lipo-miR-451a: *14-3-3ζ* significantly increased when treated with and without hemes. (**B**) Lipo-let-7i-5p: *14-3-3ζ* significantly increased when treated with and without hemes. Our treatment has been shown to increase mRNA expression of *14-3-3ζ* and, by association, the expression of heme scavengers.

**Figure 4 ijms-24-16934-f004:**
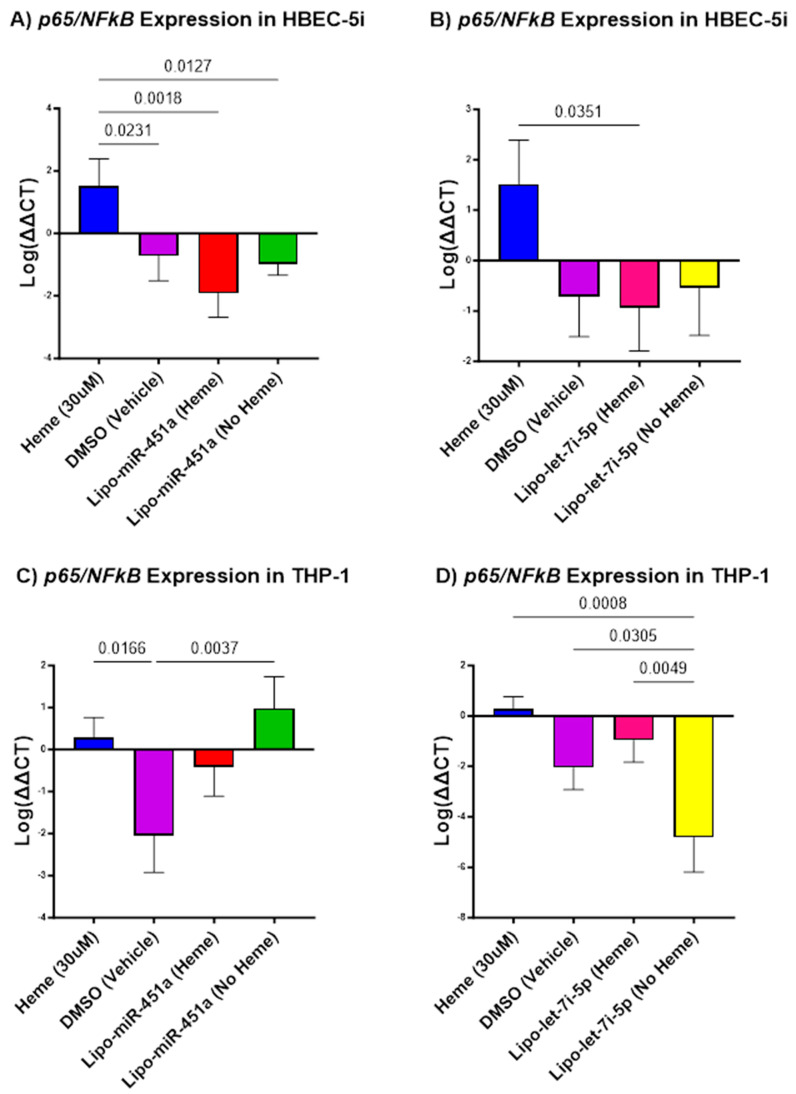
*p65*/*NFκB* mRNA expression in HBEC-5i (**A**,**B**) and THP-1 (**C**,**D**) cell culture. (**A**) Lipo-miR-451a treatment: *p65*/*NFκB* transcription significantly reduced when treated with liposomes with and without hemes compared to the heme-only treatment. (**B**) Lipo-let-7i-5p treatment: *p65*/*NFκB* expression significantly reduced when treated with liposomes with hemes compared to the heme-only group. (**C**) Lipo-miR-451a treatment: *p65*/*NFκB* expression decreased with hemes and increased when treated with liposomes without hemes. (**D**) Lipo-let-7i-5p treatment: *p65*/*NFκB* expression significantly reduced when treated with liposomes with and without hemes compared to the heme-only treatment group.

**Figure 5 ijms-24-16934-f005:**
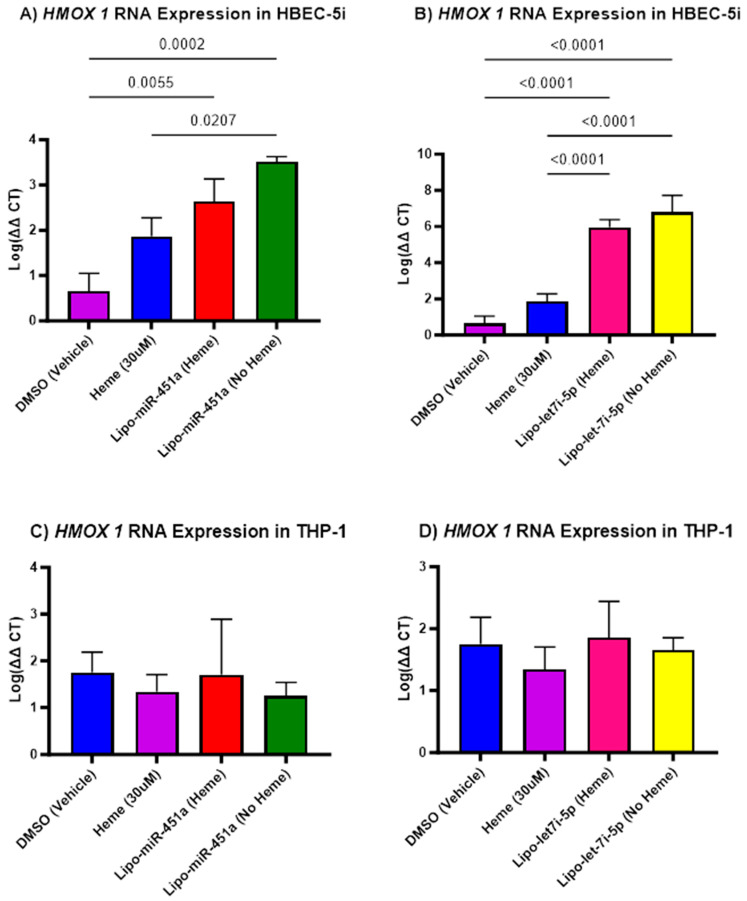
*HMOX1* mRNA expression in HBEC-5i (**A**,**B**) and THP-1 (**C**,**D**) cell culture. (**A**) Lipo-miR-451a: *HMOX1* significantly increased when treated with liposomes with and without hemes. (**B**) Lipo-let-7i-5p: *HMOX1* significantly increased when treated with liposomes with and without hemes. (**C**) Lipo-miR-451a: There is no significant increase or decrease in *HMOX1* mRNA expression. (**D**) Lipo-let-7i-5p: There is no significant increase or decrease in *HMOX1* mRNA expression. Our treatment has been shown to increase the mRNA expression of *HMOX1* in HBEC-5i cells and, by association, the expression of heme scavengers in vascular endothelial cells.

**Figure 6 ijms-24-16934-f006:**
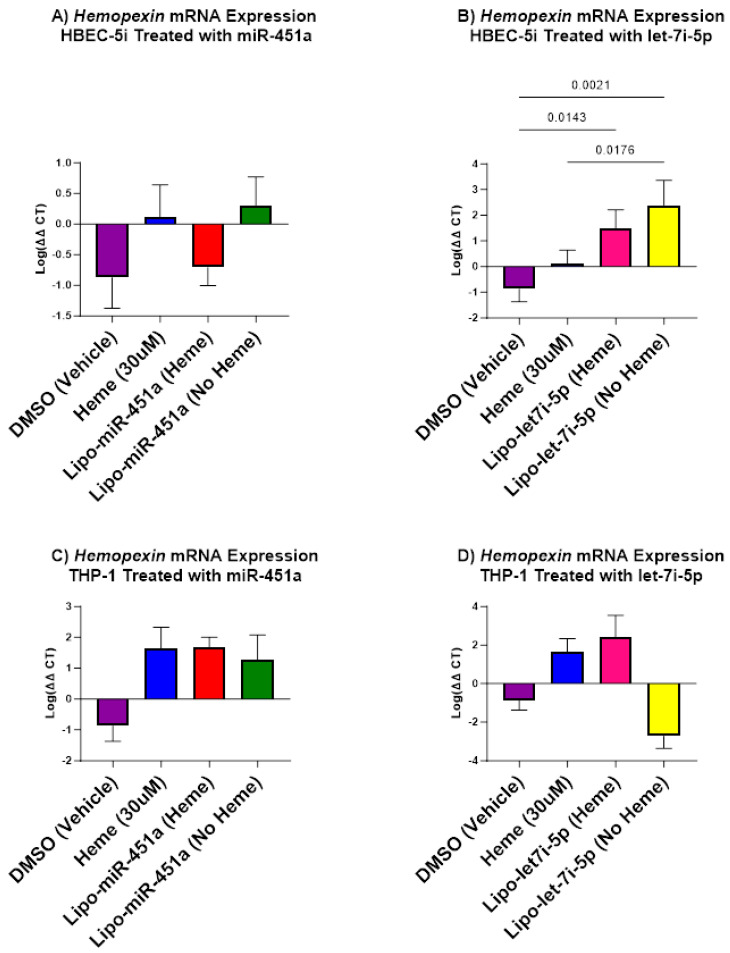
*Hemopexin* mRNA expression in HBEC-5i (**A**,**B**) and THP-1 (**C**,**D**) cell culture. (**A**) Lipo-miR-451a: There is no significant increase or decrease in *hemopexin* mRNA expression. (**B**) Lipo-let-7i-5p: *Hemopexin* significantly increased when treated with liposomes with and without hemes. (**C**) Lipo-miR-451a: There is no significant increase or decrease in *hemopexin* mRNA expression. (**D**) Lipo-let-7i-5p: *Hemopexin* significantly increased when treated with hemes and significantly decreased without hemes. Our treatment with Lipo-let-7i-5p has been shown to increase the mRNA expression of *hemopexin* in HBEC-5i cells and THP-1 cells and, by association, the expression of heme scavengers in vascular endothelial cells and macrophages.

**Figure 7 ijms-24-16934-f007:**
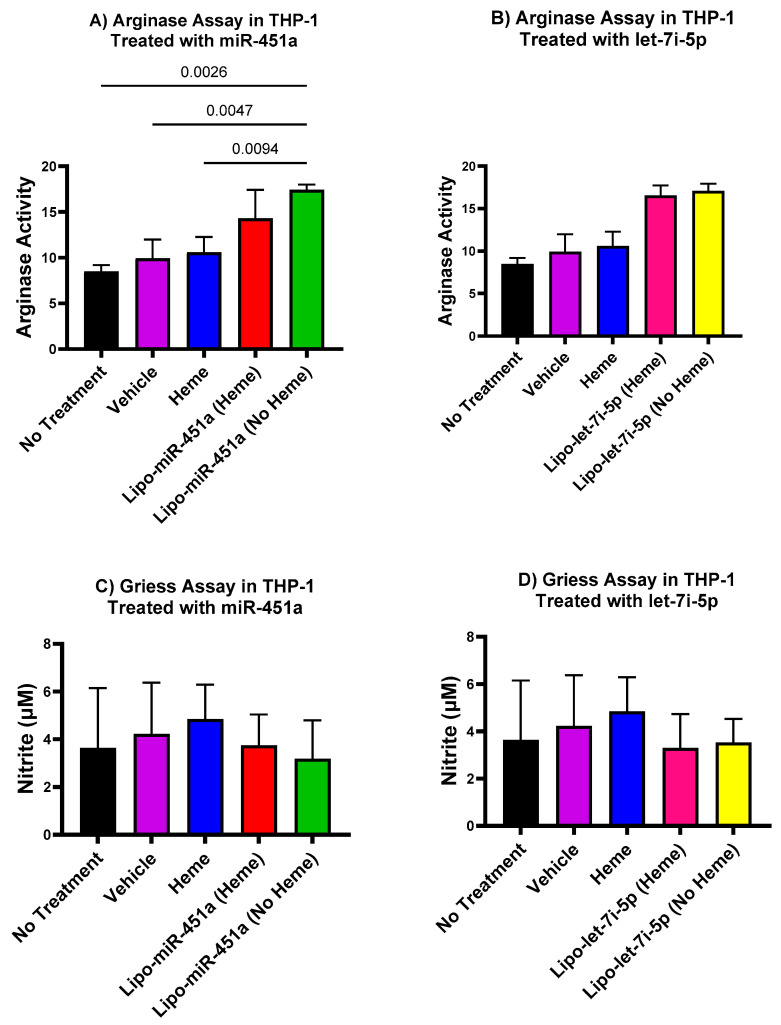
Modulation of arginase activity and nitric oxide levels by miR-451a and let-7i-5p in THP-1 cells. (**A**) Treating THP-1 cells with miR-451a significantly boosts arginase activity compared to the group exposed only to the vehicle. (**B**) A noticeable, albeit statistically non-significant, upswing in arginase activity within THP-1 cells treated with let-7i-5p. (**C**,**D**) A non-significant tendency toward reduced nitric oxide levels in THP-1 cells after treatment with miR-451a (**C**) and let-7i-5p (**D**). These results indicate that treatment with both miR-451a and let-7i-5p has the potential to induce an anti-inflammatory M2 phenotype of THP-1 cells.

**Figure 8 ijms-24-16934-f008:**
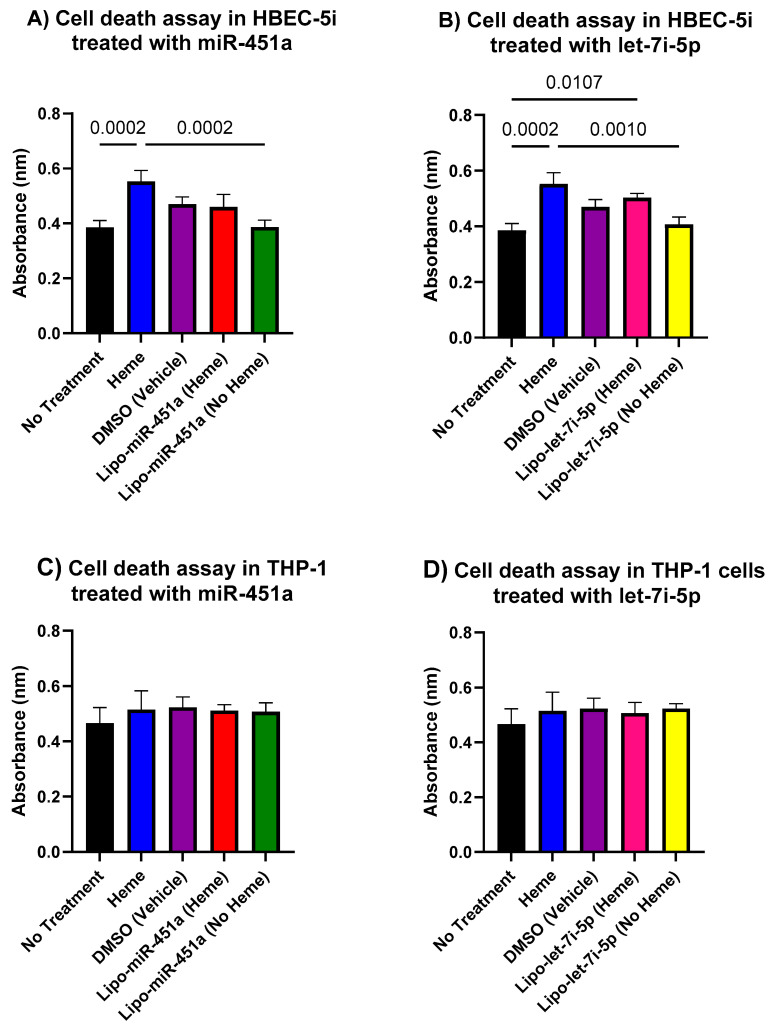
In Vitro cell death assay. Cell viability is directly proportional to optical density. (**A**) Comparing the control group to the treatment with the heme only group, there was a significant (*p* = 0.0002) decrease in absorbance. This suggests a decrease in cell viability. There is also a significant decrease in absorbance in groups treated with Lipo-miR-451a only, which means there are fewer HBEC-5i cells compared to the treatment group with hemes only. (**B**) There is a significant (*p* = 0.0010) decrease in absorbance in the Lipo-let-7i-5p treatment group, which means less cell viability with no hemes compared to the heme-only treatment group. However, there is a significant (*p* = 0.0107) increase in absorbance in groups treated with Lipo-let-71-5p. This suggests there is more cell viability compared to the no-treatment group. (**C**) The graph demonstrates no significant cell death in THP-1 cells treated with Lipo-miR-451a compared to all other groups. (**D**) There is no significant cell death in THP-1 cells treated with Lipo-let-7i-5p.

**Figure 9 ijms-24-16934-f009:**
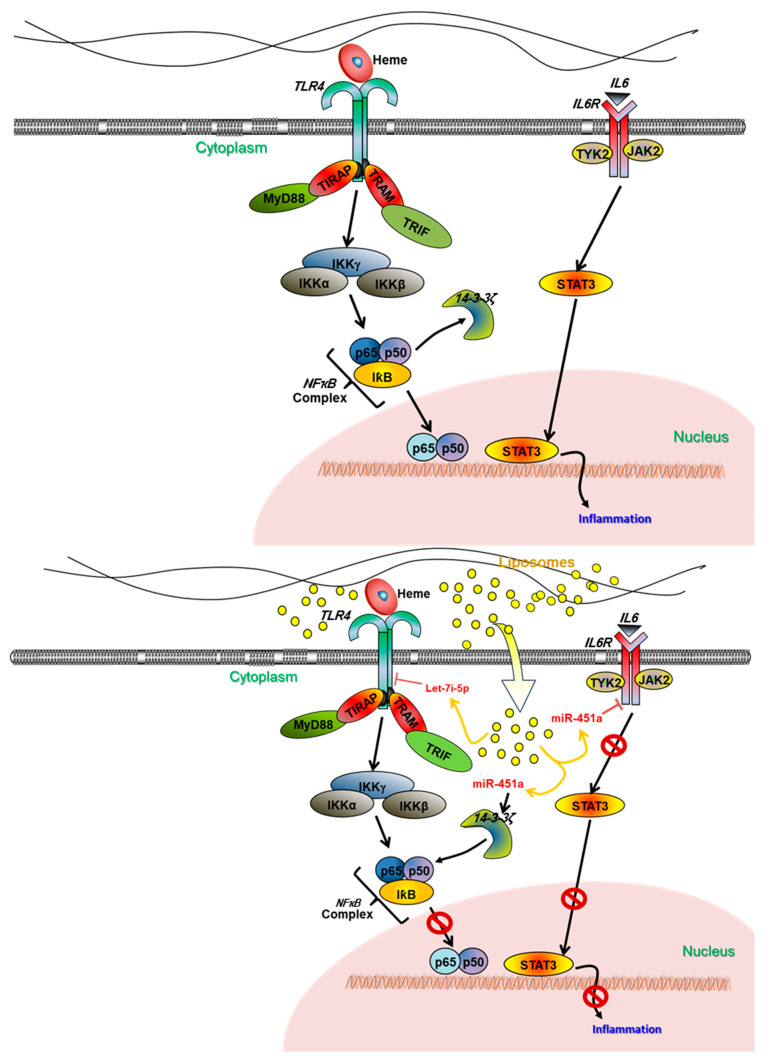
Proposed model of the molecular pathway involving *P65*/P50 (*NFκB*) transcription and *IL-6R* and *TLR4* receptor activity. (**Top**) Through two distinct routes, the activation of *TLR4* and *IL-6R* receptors in the presence of hemes or *IL-6* increases the transcription of *NFκB* and inflammation through the IKK and STAT3 pathways. (**Bottom**) Inflammation decreased due to reduced *IL-6R*, *TLR4*, and *NFκB* expression when cells are treated with encapsulated miR-451a and let-7i-5p liposomes. Due to the presence of miR-451a, there is increased *14-3-3ζ* transcription, which binds to the *NFκB* complex and prevents translocation into the nucleus.

## Data Availability

The original contributions presented in this study are included in the article/Appendix A. Further inquiries can be directed to the corresponding author.

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
