# Peer review of "Modulation of Heme-Induced Inflammation Using MicroRNA-Loaded Liposomes: Implications for Hemolytic Disorders Such as Malaria and Sickle Cell Disease"

_ijms, 2023, doi:10.3390/ijms242316934_

Round 1

Reviewer 1 Report

Comments and Suggestions for Authors

This manuscript by Bashi et al. provides original data regarding the impact of miRNA-containing liposomes on inflammation-related parameters. The results are interesting and have the potential to set the basis for the pharmaceutical use of such vehicles to ameliorate heme-induced inflammation. The manuscript will be a good addition to the selected journal and will grasp the interest of its readership. Nonetheless, I have some comments that could improve the manuscript before its acceptance.

1. Results section: The graphs and the font used on them are quite small and hard to read. Could the authors enlarge them? Especially the ones that contain 4 panels may need some rearrangement to be more easily readable.\

2. Results section: The authors don't mention in their results the outcome of arginase assay. Could they add some sentences about Figure 7?

3. Discussion section: I believe that a great part of the Discussion section is a repetition of the Results and the study's rationale. I strongly believe that it would be beneficial for the manuscript to alter this part, to focus on the explanation and discussion of their findings in the context of the current bibliography and the future potentials/perspectives. Instead of dedicating a paragraph to each finding, the authors could merge their results and discuss them altogether (when possible) to draw the complete picture, especially since they also propose a beautiful graphically presented model.

4. Discussion Section: Please enlarge the really informative graphical model to facilitate the reader.

5. I believe the reference to Supplemental Figure 1 should be moved from the Discussion Section to the Results Section since the cell viability assay results are important and need to be mentioned in the Results section to notify the reader regarding toxicity issues.

Comments on the Quality of English Language

There are some syntax and grammar errors throughout the manuscript, therefore it should be carefully proofread.

Author Response

Reviewer 1:

This manuscript by Bashi et al. provides original data regarding the impact of miRNA-containing liposomes on inflammation-related parameters. The results are interesting and have the potential to set the basis for the pharmaceutical use of such vehicles to ameliorate heme-induced inflammation. The manuscript will be a good addition to the selected journal and will grasp the interest of its readership. Nonetheless, I have some comments that could improve the manuscript before its acceptance.

  1. Results section: The graphs and the font used on them are quite small and hard to read. Could the authors enlarge them? Especially the ones that contain 4 panels may need some rearrangement to be more easily readable.
    1. We have enlarged the fonts and transformed each figure in the article from 1:4 to 2:2 panels to help with reader accessibility.

  1. Results section: The authors don't mention in their results the outcome of arginase assay. Could they add some sentences about Figure 7?

    1. In the “Results” section of the manuscript, we added, “In THP-1 derived macrophages, we observed an increase in arginase activity when treated with Lipo-miR-451a. This observation is interesting as increased arginase levels indicate a trend to an M2 anti-inflammatory phenotype. A similar tendency toward elevated arginase activity was observed in THP-1 derived macrophages exposed to Lipo-let-7i-5p treatment, albeit without reaching statistical significance (Figure 7B). We also detected a notable downward trend (without significance) in nitrite levels in THP-1-derived macrophages treated with both Lipo-miR-451a and Lipo-let-7i-5p (Figure 7C and 7D).” This can be highlighted in the manuscript and can be found on lines 416-423. We also discuss the arginase data under the “Discussion” header on lines 546-548 and 551-553. On these lines, we say, “Our findings, as depicted in Figure 7, effectively demonstrate the potential utility of Lipo-miR-451a or Lipo-let-7i-5p in promoting the augmentation of M2 macrophage phenotypes. […] Also, in Figure 7, we successfully demonstrated an elevation in arginase levels alongside a reduction in nitrite, aligning with the characteristic pattern of di-minished inflammation.”

  1. Discussion section: I believe that a great part of the Discussion section is a repetition of the Results and the study's rationale. I strongly believe that it would be beneficial for the manuscript to alter this part, to focus on the explanation and discussion of their findings in the context of the current bibliography and the future potentials/perspectives. Instead of dedicating a paragraph to each finding, the authors could merge their results and discuss them altogether (when possible) to draw the complete picture, especially since they also propose a beautiful graphically presented model.

    1. Thank you for these comments. n the discussion section, we endeavored to rephrase content, minimizing result repetition unless essential to enhance the discussion's coherence. To help facilitate understanding, we have edited and revised the discussion to include future potentials and perspectives. We have altered lines 505-507:” Inflammatory conditions are regulated by a balance of anti-and pro-inflammatory cytokines and chemokines. 14-3-3ζ, an important anti-inflammatory component, showed interesting results in our studies.”  516-520: “IL-6R, TLR4, and 14-3-3ζ are all known to impact the transcription of P65/NFÒ¡B directly and indirectly 27-29. These facts are supported by our results in Figure 4 further confirming the role of Lipo-miR-451a or Lipo-let-7i-5p in regulating inflammatory responses and their potential application as a therapy for heme-induced inflammation.” 521-523: “Given the role of chronic inflammation due to cytotoxic heme, heme scavengers such as HO-1 were considered as potential targets for the miRNA treatment.”  546-548: “Our findings, as depicted in Figure 7, effectively demonstrate the potential utility of Lipo-miR-451a or Lipo-let-7i-5p in promoting the augmentation of M2 macrophage phenotypes.” 551-553: “[…] and supports our hypothesis.” and 596-598: “Here, we observed a proinflammatory phenotype for NFÒ¡B in the presence of excessive heme. However, this inflammatory response can be mitigated by miRNAs miR-451a and let-7i-5p (Figure 9).” We also included a paragraph describing figure 8 which summarizes the molecular pathways mentioned in the manuscript. The paragraph can be found on lines 558-566. On these lines, we state: “We explored the role of cytotoxic heme in triggering chronic inflammation and further investigated the potential role of miRNA-loaded liposomes as a novel therapeutic strategy to mitigate pro-inflammatory responses caused by free heme. Our results demonstrated that Lipo-miR-451a and Lipo-let-7i-5p, effectively regulated pro-inflammatory pathways, reducing expression of p65/NFÒ¡B and pro-inflammatory cytokines in both HBEC-5i and THP-1 cells while remaining relatively non-toxic to the cells (Figure 8). This suggests that our treatment can modulate these molecular pathways and produce an anti-inflammatory environment in those where cytotoxic heme present.  ”

  1. Discussion Section: Please enlarge the really informative graphical model to facilitate the reader.
    1. We have enlarged Figure 9 to facilitate reader comprehension.

  1. I believe the reference to Supplemental Figure 1 should be moved from the Discussion Section to the Results Section since the cell viability assay results are important and need to be mentioned in the Results section to notify the reader regarding toxicity issues.
    1. We moved Supplemental Figure 1 to the discussion section as Figure 8. We included the figure legend on lines 459-470: “Figure 8: In vitro cell death assay. Cell viability is directly proportional to the optical density (A) Comparing the control group to the treatment with heme only group, there was a significant (p=0.0002) decrease in absorbance. This suggests a decrease in cell viability. There is also a significant decrease in absorbance in groups treated with Lipo-miR-451a only, which means there are less HBEC-5i cells compared to the treatment group with heme only. (B) There is a significant (P=0.0010) decrease in absorbance in Lipo-let-7i-5p treatment group which means less cell viability with no heme com-pared to the heme only treatment group. However, there is a significant (P=0.0107) increase in absorbance in groups treated with Lipo-let-71-5p. This suggests there is more cell viability compared to the no treatment group. (C) The graph demonstrates no significant cell death in THP-1 cells treated with Lipo-miR-451a compared to all other groups. (D) There is no significant cell death in THP-1 cells treated with Lipo-let-7i-5p.” We also included a brief summary of the results on lines 472-474: “Figure 8 shows in vitro cell death assay data, it appears that our model exhibits a greater degree of stability and lower toxicity in THP-1 macrophages compared to HBEC-5i endothelial cells.” We also included information in the discussion of the same figure on lines 552-554: “Also, in Figure 7, we successfully demonstrated an elevation in arginase levels alongside a reduction in nitrite, aligning with the characteristic pattern of diminished inflammation.”

  1. There are some syntax and grammar errors throughout the manuscript, therefore it should be carefully proofread.

    1. We have read and revised the paper to address any syntax or grammatical errors that may be present.

Reviewer 2 Report

Comments and Suggestions for Authors

Two cell models, endothelial cells (HBEC-5i) and macrophages (THP-1), were applied to evaluate effects of selected micro RNAs on heme-induced inflammatory response. Indeed, both miR-451a and let-7i-5p reduced heme-mediated inflammation. Expression of hemopexin and heme oxygenase 1 increased in both cell lines, and a switch to M2-type macrophages occurred in THP-1 cells. These results are important for further testing of liposomal encapsulated micro RNAs for therapeutic approaches against malaria and sickle cell disease.

I have only one minor remarks.

Line 278:             that is doubled

Author Response

Reviewer 2:

Two cell models, endothelial cells (HBEC-5i) and macrophages (THP-1), were applied to evaluate effects of selected micro RNAs on heme-induced inflammatory response. Indeed, both miR-451a and let-7i-5p reduced heme-mediated inflammation. Expression of hemopexin and heme oxygenase 1 increased in both cell lines, and a switch to M2-type macrophages occurred in THP-1 cells. These results are important for further testing of liposomal encapsulated micro RNAs for therapeutic approaches against malaria and sickle cell disease.

  1. I have only one minor remarks. Line 278: that is doubled

  1. Thank you for your comment. To address this, we removed the repetitiveness in the discussion and have reworded our paragraph as an interpretation of our results. This can be found on lines 517-521: “IL-6R, TLR4, and 14-3-3ζ are all known to impact the transcription of P65/NFÒ¡B directly and indirectly 27-29. These facts are supported by our results in Figure 4 further confirming the role of Lipo-miR-451a or Lipo-let-7i-5p in regulating inflammatory responses and their potential application as a therapy for heme-induced inflammation.”

Round 2

Reviewer 1 Report

Comments and Suggestions for Authors

The authors have addressed all my concerns and I believe their article can be published as it is and will be of interest to the readership of the journal.